# Mask-Guided Spatial–Spectral MLP Network for High-Resolution Hyperspectral Image Reconstruction

**DOI:** 10.3390/s24227362

**Published:** 2024-11-18

**Authors:** Xian-Hua Han, Jian Wang, Yen-Wei Chen

**Affiliations:** 1Graduate School of Artificial Intelligence and Science, Rikkyo University, Tokyo 171-8501, Japan; 2School of Information Science and Engineering, Shandong Normal University, Jinan 250358, China; jwang@sdnu.edu.cn; 3College of Information Science and Engineering, Ritsumeikan University, Osaka 603-8577, Japan; chen@is.ritsumei.ac.jp

**Keywords:** hyperspectral image reconstruction, degradation, sensing mask, spatial–spectral modelling, long dependency, MLP network

## Abstract

Hyperspectral image (HSI) reconstruction is a critical and indispensable step in spectral compressive imaging (CASSI) systems and directly affects our ability to capture high-quality images in dynamic environments. Recent research has increasingly focused on deep unfolding frameworks for HSI reconstruction, showing notable progress. However, these approaches have to break the optimization task into two sub-problems, solving them iteratively over multiple stages, which leads to large models and high computational overheads. This study presents a simple yet effective method that passes the degradation information (sensing mask) through a deep learning network to disentangle the degradation and the latent target’s representations. Specifically, we design a lightweight MLP block to capture non-local similarities and long-range dependencies across both spatial and spectral domains, and investigate an attention-based mask modelling module to achieve the spatial–spectral-adaptive degradation representationthat is fed to the MLP-based network. To enhance the information flow between MLP blocks, we introduce a multi-level fusion module and apply reconstruction heads to different MLP features for deeper supervision. Additionally, we combine the projection loss from compressive measurements with reconstruction loss to create a dual-domain loss, ensuring consistent optical detection during HS reconstruction. Experiments on benchmark HS datasets show that our method outperforms state-of-the-art approaches in terms of both reconstruction accuracy and efficiency, reducing computational and memory costs.

## 1. Introduction

Hyperspectral images (HSIs) contain many more spectral bands than conventional RGB images, allowing them to capture richer details. Because of their abundant spectral information, HSIs are extensively used in various applications, such as image recognition [1,2,3], remote sensing [4,5,6,7], medical image analysis [8,9], and environmental monitoring. Traditionally, HSIs are acquired using spectrometers that scan scenes across their spectral or spatial dimensions. The scan process is usually time-consuming and unsuitable for capturing dynamic objects. To address this limitation, snapshot compressive imaging (SCI) systems [10,11,12,13] have been developed to capture HSIs at video rates. Among these, coded aperture snapshot spectral imaging (CASSI) [14,15,16] stands out as an promising imaging modality for capturing high-quality spatial and spectral details in real time. In general, CASSI uses a coded aperture and a disperser to modulate HSI signals at different wavelengths, compressing them into a 2D measurement. THe reconstruction of the full HSI from the compressed data, known as the CASSI inverse problem, is a challenging task due to its ill-posed nature. To address this challenge, various approaches, primarily ranging from traditional model-based methods [17,18,19,20,21] and end-to-end (E2E) techniques [22,23,24,25,26,27,28,29] to deep unrolling pipelines [25,30,31,32,33], have been proposed, resulting in substantial advancements.

Traditional model-based approaches typically formulate the compressed detection process as a mathematical model and utilize various handcrafted priors, such as total variation [34,35], low ranks [17], sparsity [36,37], and many others [38,39], to enhance reconstruction accuracy. Although these techniques enable practical spectral reconstruction, they come with the drawback of high computational costs and require the manual design of priors for each specific scene, limiting their broader applicability in real-world systems. Inspired by the remarkable success of deep learning in vision tasks, researchers have extensively explored the use of convolutional neural networks (CNNs) [23,24,26,40] and transformers [28,41,42] for hyperspectral image reconstruction. These deep learning methods are generally implemented in an end-to-end (E2E) learning way to establish brute-force mapping between compressed measurements and the target hyperspectral images [23,24,29,40,42], and they are limited in their interpretability as black-box models. Recent research has increasingly turned to deep unfolding networks (DUNs) as a promising hybrid approach, aiming to combine the strengths of both model-based and deep learning techniques [25,30,31,32]. DUNs decompose the whole optimization process into two distinct sub-problems and are usually realised in a multi-stage iterative way, which unavoidably causes a large computational cost and memory footprint.

In addition, the backbone architectures of the E2E and DUN method for HSI reconstruction have progressed from CNNs [22,23,24] to transformers [31,32,42]. Benefiting from the powerful modelling capability of their long-range dependencies, transformers have demonstrated promising performances. To reduce computational complexity and effectively model spectral correlations, transformer-based HSI models typically focus on self-attention (SA) mechanisms along the spectral dimension [31,42], rather than in the spatial domain. This spectral SA modelling method generally uses a few conventional convolution layers to exploit local spatial dependencies. However, while leveraging non-local similarities and long-range relationships in the spatial domain can be highly beneficial for recovering missing information from HSIs, it usually leads to a high computational cost and large memory overhead. Despite the efficiency of improvements made with window-based transformers [43], the computational load for real imaging sensors remains a challenging aspect that needs to be reduced further. In contrast, recent developments have shown that replacing the self-attention mechanism in transformers with MLP-like operations offers a promising alternative. These operations excel at capturing non-local similarities and modelling long-range dependencies, with considerable potential applicability in the vision community. Notably, they have delivered comparable performance in various vision tasks while reducing memory usage and computational costs. Building on the above insights, we explore the use of the MLP block as the core element for constructing a baseline network for hyperspectral (HS) image reconstruction and aim to offer a compelling solution to overcome the locality constraints of CNNs and the high computational demands of Vision Transformers (ViTs).

To this end, the study introduces a novel spatial and spectral MLP (S2MLP) block, designed to effectively capture long-distance representations from both non-local spatial positions and spectral bands. This proposed S2MLP block serves as the core component used to construct a simple yet effective deep model (S2MLPNet) for HSI reconstruction. In particular, the spatial MLP in the S2MLP block replaces the conventional SpatialFC with a CycleFC layer [44], which maintains a comparable linear computational cost to that of the ChannelFC while enhancing non-local spatial modelling. Additionally, the S2MLPNet integrates a multi-level information fusion module, which promotes the flow of valuable information between different MLP blocks. Moreover, we equip the reconstruction heads with multiple S2MLP features for deep supervision at various stages, facilitating the effective representation learning of the overall model. Then, an attention-based mask modelling module has been investigated to achieve the spatial–spectral-adaptive degradation representation fed to the S2MLPNet, disentangling the representation learning of the latent HSI and the degradation information. Finally, a dual-domain loss function, combining losses from both the compressive and reconstruction domains, is introduced to ensure consistency in optical detection. Extensive experiments on benchmark datasets demonstrate that our proposed method significantly outperforms existing state-of-the-art (SOTA) approaches.

In summary, our main contributions are as follows:(1)We propose a simple yet effective S2MLPNet for hyperspectral image reconstruction. As far as we know, this study marks the first effort to investigate the potential of MLPs in HSI reconstruction. We believe it will offer new perspectives for future research on deep modelling architectures for HSIs.(2)We design a lightweight spatial and spectral MLP (S2MLP) module to efficiently capture non-local similarities and long-range dependencies across both spatial and spectral domains with linear complexity to the image size.(3)We investigate the use of an attention-based mask modelling module to capture the degradation representation and incorporate it into the S2MLP for learning the disentangled representations of latent HSI data.(4)We exploit the multi-level information fusion mechanism seen between different levels and the deep supervision strategy used to enhance robust representation learning and further employ a dual-domain loss function for ensuring optical measurement consistency.

## 2. Related Work

This section firstly presents a brief review of the incorporation strategies of the degradation mask used for the HSI reconstruction and then presents the network architectures employed in recent learning-based methods, including E2E and deep unrolling approaches.

### 2.1. Incorporation of Degradation into HSI Reconstruction

A spectral compressive imaging system such as CASSI captures 3D spatial and spectral data as a 2D measurement, where the detection process can be expressed as a mathematical model by applying the degradation mask to the original HSIs. Traditional model-based methods [17,18,19,20,21] minimize the regularised objective function formulated based on the degradation process and the handcrafted priors of the latent HSIs. Then, optimization of the model-based methods iteratively achieves the recovery HSI by an inverse projection of the previous reconstruction error according to the degradation mask. Although this kind of method makes full use of the physical degradation process and its clear interpretability, it usually suffer from an insufficient representation ability due to the challenge of creating prior designs for each specific scene. Deep neural networks have been extensively applied for HSI reconstruction to achieve a brute-force mapping (E2E methods) between the measurement and the target HSI while usually ignoring the potential degradation information found in the detection process [22,23,24,40]. A few works have attempted to integrate the degradation mask into a representation learning network to enhance modelling capabilities [29,42] and have demonstrated some performance enhancements in HSI reconstruction. Recently, the deep unrolling method (DUM) [25,30,31,32] has been proposed to integrate the degradation process-formulated model method and the deep learning approach. Benefiting from the advantages of two pipelines, DUM methods usually have great potential for HSI reconstruction and demonstrate significant performance gains [31,32,33,45]. For example, DGSMP [31] employed an iterative approach that combines Maximum A Posteriori (MAP) estimation with a learnable Gaussian Scale Mixture prior. DAUHST [32] introduced a degradation-aware framework and half-shuffle attention that incorporated a prior learning network to tackle issues of degradation and ill-posedness. RDLUF [45] proposed a residual degradation learning unfolding framework to bridge the gap between the sensing matrix and the degradation process. Despite the great progress made in HSI reconstruction, DUM is generally implemented in a multi-stage, iterative way and thus has a high computational cost and heavy memory footprint, restricting its wide applicability in real imaging systems.

### 2.2. Network Architectures for HSI Representation Learning

The network architectures used for HSI representation learning have evolved from convolution-based to transformer-based backbones [23,24,26,28,31,32,40,42]. Initially, CNN-based approaches were exploited to capture local spatial similarities in HSI data. For instance, λ-net [24] proposed a dual-branch convolution network to reconstruct HSIs, while TSA-Net [22] is designed to model spatial–spectral correlations via elaborating various combinations of convolution operations. Further, HypermixNet [26] explored improved variants of the convolution to enhance its representation ability as well as reduce computational complexity. Although CNNs are effective to some extent, their inherent inductive biases usually limit their ability to capture non-local similarities. To overcome this, transformer-based methods have emerged [41,42], which are used to employ multi-head self-attention mechanisms to capture long-range dependencies. For example, MST [42] investigated a the use of a spectral transformer via capturing dependencies across spectral dimensions, generating attention maps that implicitly encode global context. CST [41] incorporated spatially sparse nature into a transformer learning model to capture the interactions of closely related patterns and reduce computational complexity. PADUT [33] investigated the use of a non-local spectral transformer to model the representative 3D features of HSIs. More recently, several studies [46,47] have integrated the window-based spectral transformer with the spatial modelling module to capture both spatial and spectral correlations and demonstrated promising reconstruction performances. All of these transformer-based networks generally realize long-region dependencies by calculating the pair-wise correlations of all tokens as a self-attention map, which would unavoidably increase the computational complexity with the growth of the token number.

In recent years, a novel type of deep learning architecture built entirely on multi-layer perceptrons (MLPs) has gained significant interest in the computer vision domain. For example, MLPMixer [48] pioneered this shift by introducing the concept of MLP-based prototypes, while GMLP [49] advanced the idea further by incorporating gating mechanisms to boost performance. Additionally, ResMLP [50] made key improvements by adding residual connections, which enhanced its training stability. Contrarily, CycleMLP [44] tackled the persistent challenge of the computational reliance on the input sizes from an earlier transformer era. Although MLP-based models have proven effective in various high-level tasks, their application in low-level tasks remains underexplored. MLP-based networks have promising prospects in capturing long-range dependencies as they separately mix tokens and channels across layers, which is significantly different from the CNNs used for local representation learning and ViTs used for global dependency modelling with self-attention (SA) computation. Therefore, compared to CNNs and transformers with an SA mechanism, MLP-based models generally have lower computational complexity and are particularly suitable for low-level HSI reconstruction tasks that use high-dimensional information, where non-local information is essential for restoring missing or corrupted data. The investigation of MLP models will offer new perspectives on deep modelling architectures, opening up more opportunities for diverse and adaptable network configurations specifically in high-dimensional data representations.

### 2.3. The Coded Mask in Spectral Snapshot Imaging Systems

The coded mask in spectral snapshot imaging systems allows for the efficient capture of spectral data by encoding both spatial and spectral information into a single image, which can be later reconstructed to provide a full hyperspectral cube. One of the most popular systems in snapshot compressive imaging (SCI) is the Coded Aperture Snapshot Spectral Compressive Imaging (CASSI) system [14,15,16], which usually uses the same coded mask to encode different spectral bands and spatially shifted the coded spectral images with a disperser to integrate them into a single snapshot. In the early era of this research, most works usually randomly generated a binary matrix according to a Bernoulli distribution, with *p* = 0.5 as the coded masks [25,26,28], and then simulated the snapshot measurements for HS image reconstruction. Meng et al. [22] have released a coded mask obtained with optical hardware, and then subsequent studies have widely employed it to conduct simulation experiments for fair comparisons [23,24,31,32,42]. More recently, Motoki et al. [51] proposed a novel optical filter-based HS imaging system using a spatial–spectral coded 3D mask to capture compressive measurements, and released a coded mask. This study uses a 3D coded mask in its simulation experiments.

## 3. Spectral Snapshot Imaging Model

According to compressive sensing theory [52], a spectral snapshot imaging system captures a compressed measurement by encoding information from all spectral bands. Figure 1 provides an overview of the basic coding process. Given a three-dimensional (3D) spectral image $X\in {ℜ}^{H\times W\times \Lambda }$ of a scene, where *H* and *W* represent the height and width, while Λ refers to the wavelength number, the spectral snapshot imaging system employs a coded aperture (sensing mask) $M_{\lambda }\in {ℜ}^{H\times W}$ to modulate the image of each spectral band, and the resulted temporary data $Y_{\lambda }^{\prime }\in {ℜ}^{H\times W}$ can be expressed as
(1)$$Y_{\lambda }^{\prime }==X\left(:,:,\lambda \right)⊙M_{\lambda },$$
where ⊙ represents the element-wise multiplication and $Y^{\prime }\in {ℜ}^{H\times W\times \Lambda }$ refers to the modulated cube. Then, $Y^{\prime }$ is shifted in the horizontal direction using a dispersive function *d* (representing the spatial shifting step), resulting in the dispersed cube $Y^{\prime \prime }\in {ℜ}^{H\times (W+d(\Lambda -1))\times \Lambda }$, which can be described as follows:(2)$$Y^{\prime \prime }\left(w,h,\lambda \right)=Y^{\prime }\left(w,h+d\left(\lambda \right),\lambda \right),$$
where *h* and *w* refer to the spatial coordinates. Ultimately, the compressive measurement Y is achieved by performing spectral integration on the dispersed cube $Y^{\prime \prime }$, which is expressed as follows:(3)$$Y=\sum_{\lambda =1}^{\Lambda }Y^{\prime \prime }\left(:,:,\lambda \right)+N,$$
where N denotes the random noise generated during the measurement process. To simplify, we rewrite Equation (Equation 3) in a matrix–vector format as follows:(4)y=Φx+n,
where Φ refers to the sensing matrix, characterised by its large and highly structured form. In particular, the CASSI system employs the same sensing mask for different spectral bands and uses a disperser by spatially shifting the modulated cube used for integration. While the recent work [51] adopted an optimised spatial and spectral coded mask, which can be easily achieved by placing the encoding (Fabry–P$\overset{´}{e}$rot) filter, along with an array CMOS devices, onto a monochromatic image sensor to encode the original HS cube and integrating the spectral information into snapshot measurements without spatial shift on the spectral bands. This novel spectral compressive imaging is usually referred to as an optical filter-based HS imaging system. This study obtains snapshot images using the optical filter-based HS imaging technique. After the compressive spectral measurement, it is an indispensable step to recover the 3D HSI x from the compressed 2D measurement y, which is a challenging task due to its ill-posed nature. Typically, the sensing mask varies spatially and causes different information losses in different spatial positions and spectral bands. Therefore, the incorporation of the sensing matrix into inverse HSI reconstruction is crucial to obtain a disentangled representation of the latent HSI while weakening the effect of the degradation.

## 4. Proposed Method

With the measured snapshot Y and the corresponding sensing mask Φ, the goal of HSI reconstruction is to accurately recover the underlying 3D cube X. The challenge lies in reversing the effects of the sensing process to retrieve high-dimensional spectral information from the compressed measurement. This study introduces a simple yet highly effective deep learning model for HSI reconstruction, as depicted in Figure 2. The architecture is composed of two distinct branches. The first branch is designed to capture both spatial and spectral information, utilising a series of mask-guided spatial and spectral MLP blocks (MG-S2MLP), to learn the complicated and abundant structures in high-dimensional HS data. The second branch focuses on mask modelling and incorporates several specialized modules that are integrated with each MG-S2MLP block. Furthermore, an interactive mechanism is established between the two branches, allowing for cross-branch communication, as well as interactions between the multi-level MG-S2MLP blocks, to enhance the reconstruction performance. This dual-branch scheme ensures that both spatial–spectral features and mask properties are jointly optimized throughout the reconstruction process.

Specifically, the snapshot measurement Y and the sensing mask Φ are separately passed through a convolutional layer, producing the initial representations $X_{0}$ for the measurement and $M_{0}$ for the mask, both with a channel dimension of *C*. To enhance the mask representation, an attention-based mask modelling module (AMMM) is applied, generating a mask context, $X_{0}^{M}=f_{MAMM}^{0}\left(M_{0}\right)$, which interacts with the measurement representation $X_{0}$ of the S2MLP branch through a multiplication operation. Then, after the linear mapping of the aggregated measurement and mask context, we add $X_{0}$ to enhance its representation capability. The resulting representation is then fed into a S2MLP module, which is designed to model both spatial and spectral dependencies simultaneously. The above process, referred to as the MG-S2MLP module, is formally represented as follows:(5)$$X_{0,1}=f_{S2MLP}\left(f_{lp}\left(X_{0}^{M}\right)+X_{0}\right).$$
where $f_{lp}\left(\cdot \right)$ denotes a linear mapping function. Next, an another MG-S2MLP module is applied, constructing a full S2MLP block at the first level, which generates spatial–spectral feature representation, denoted as $X_{0}$. This process is repeated over several levels, where the MG-S2MLP block at the l−th level produces the corresponding feature representation $X_{l}$ for that level, with l=1,2,⋯,L, where *L* is the total number of levels. To ensure that valuable information from previous levels is effectively transmitted and utilised at each subsequent stage, we introduce a multi-level information fusion module (MIFM). This module aggregates important features from all preceding levels (i.e., from levels 1,2,⋯,l−1) to form the input for the current l−th level. The MIFM strategically fuses information across multiple levels to enhance the learning of spatial and spectral dependencies, and this process is mathematically formulated as
(6)$$X_{l}^{in}=f_{PW}\left(\left[X_{1},X_{2},\cdots ,X_{l-1}\right]\right)$$
where $f_{PW}\left(\cdot \right)$ refers to a point-wise convolution for multi-level feature fusion. Once the spatial–spectral features from all *L* levels have been learned, we apply a prediction head with simple convolutional layers to the output features of each level. These prediction heads are responsible for generating the HS reconstructions at all *L* levels, producing *L* different HS predictions. These predictions are then used to formulate multiple loss functions, enabling the effective optimization of the reconstruction model. The overall losses are obtained as follows:(7)$$L=L_{L}(f_{PH}^{L}\left(X_{L}\right))+\alpha \sum_{l=1}^{L-1}L_{l}(f_{PH}^{l}(X_{l}))$$
where $f_{PH}^{l}$ refers to the prediction head applied at the l−th level, which is responsible for generating a reconstructed HSI from the learned representations $X_{l}$. The loss function $L_{*}$ at each level consists of two components: a reconstruction loss and a projection loss. The reconstruction loss measures the difference between the ground-truth HSI X and the prediction at each level, while the projection loss evaluates the difference between the snapshot measurement X and the degraded prediction using the simulated compressive sensing process $f_{CS}$. This dual-loss function ensures that our model not only reconstructs the HSI but also aligns it with the observed measurement. The loss at the l−th level can be expressed as
(8)$$L_{l}={∥X-f_{PH}^{l}(X_{l})∥}^{2}+{∥Y-f_{CS}\left(f_{PH}^{l}\left(X_{l}\right)\right)∥}^{2}$$

Next, we will present a detailed description of our proposed spatial and spectral representation leaning component, S2MLP, and the attention-based mask modelling module, AMMM.

**Spatial and spectral MLP module (S2MLP):** The S2MLP module is composed of two key components: a spatial MLP that facilitates the communication of information across the spatial domain and a spectral MLP that manages the interactions among spectral bands. To process the input feature, layer normalization (LN) is employed to standardize the input for the following exploitation of spatial correlations. Unlike the conventional Spatial Fully-Connected layer (SpatialFC) [48], which has a quadratic computational complexity related to the size of the feature map, S2MLP employs Cycle Fully-Connected layers (CycleFC) [44], offering a more efficient approach to capturing long-range dependencies with the linear complexity of number of feature channels. In detail, CycleFC achieves spatial interaction by sampling points cyclically along the channel dimension, maintaining the same computational complexity as Channel Fully-Connected layers (ChannelFC) [48], but with a significantly larger receptive field. This cyclical sampling allows CycleFC to more effectively capture spatial dependencies across the entire feature map without the large computational cost of traditional methods. In essence, S2MLP uses three parallel CycleFC layers, each with different step sizes: $S_{H}\times 1$, 1×1, and $1\times S_{W}$. The integration of these three layers captures potential spatial interactions across a receptive field of size $S_{H}\times S_{W}$, offering a broad coverage of spatial relationships within the feature map. In our experiments, $S_{W}$ and $S_{H}$ are both set to 11, as illustrated in the bottom–center panel of Figure 2. In general, the operation of CycleFC with a step size of $S_{H}\times S_{W}$ can be mathematically expressed as follows:(9)$$\begin{array}{cc} F_{S_{H}\times S_{W}}=CycleFC(F_{i,j,:}) & =\sum_{c=0}^{C_{in}}F_{i+\sigma_{i}\left(c\right),j+\sigma_{j}\left(c\right),c}W_{c,:}^{CFC}+b\\ \sigma_{i}\left(c\right) & =\left(c\;mod\;S_{H}\right)-1,\\ \sigma_{j}\left(c\right) & =\left(\frac{c}{S_{H}}\;mod\;S_{W}\right)-1, \end{array}$$
where $W^{CFC}\in R^{C_{in}\times C_{out}}$ and $b\in R^{C_{out}}$ denote the weighting matrix and bias of the CycleFC operator, respectively, and the terms $\sigma_{i}\left(c\right)$ and $\sigma_{j}\left(c\right)$ refer to the spatial offset in the horizontal and vertical directions for the c−th channel, determining how the feature map is sampled during cyclical operation. These offsets enable the CycleFC layers to efficiently capture long-range spatial dependencies by shifting across the feature map in both spatial dimensions. Subsequently, we sum the outputs from the three parallel CycleFC layers, each operating with different step sizes to capture spatial interactions at varying scales. This summation aggregates spatial information from all three receptive fields, resulting in a richer spatial representation. To further refine this representation, we apply a linear mapping function, denoted as $f_{lm}\left(\cdot \right)$, which transforms the aggregated output into the final spatial feature $F^{Spa}$. This process is formally expressed as:(10)$$F^{Spa}=f_{lm}\left(F_{11\times 1}+F_{1\times 1}+F_{1\times 11}\right)$$

Next, the spectral MLP is designed to capture interactions across the spectral domain. Similar to the spatial MLP, layer normalization is firstly employed to standardize the input features, and then a ChannelFC layer is applied to aggregate information across the channels with a step size of 1, effectively increasing the number of channels in the feature map by a factor of four. By expanding the channel dimension, the model can capture a broader range of spectral dependencies. After the channel expansion, a GeLU activation function is applied, introducing non-linearity to enhance the feature learning process. Finally, another ChannelFC layer is used to restore the channel number back to its original size, ensuring that the output retains the same dimensions as the input while preserving the spectral relationships learned during the process. This entire sequence of operations enables the spectral MLP to model complex spectral interactions efficiently, balancing both expansion and compression in the channel domain. The overall process of the spectral MLP can be mathematically expressed as follows:(11)$$F^{Spe}=f_{FC+GeLU}^{2}\left(f_{FC+GeLU}^{1}\left(F^{Spa}\right)\right)$$

**Attention-based mask modelling module:** The mask modelling branch is composed of several independent attention-based mask modelling modules (AMMMs), each aligned with a corresponding S2MLP module. These AMMMs are designed to capture the 3D spatial–spectral mask context, which is then used to modulate the input features of the S2MLP. This integrated process, referred to as MG-S2MLP, is crucial for disentangling the latent HS data from the degradation mask, thereby improving the network’s ability to generalize across different conditions and datasets. Specifically, the AMMM begins by transforming the sensing mask M through a convolutional layer, generating an initial mask $M_{0}$ with the same dimensions as the input feature of the corresponding S2MLP. This ensures that both the mask and feature map are aligned for subsequent operations. Next, a depth-wise convolution is applied to the mask feature, followed by a Sigmoid activation function, producing a mask attention map $A_{M}$. By element-wise multiplying the mask attention map $A_{M}$ and feature $M_{0}$, we obtain a modulated mask feature, and this is then passed through another convolution layer to generate the final mask context, which is subsequently used to interact with the S2MLP module. By incorporating mask-specific information, the network can better understand the relationship between the mask and the underlying HS data, leading to more accurate reconstructions. This entire process can be mathematically described as follows:(12)$$F^{mask}=M_{0}⊙f_{Sigmoid}\left(f_{Dw}\left(M_{0}\right)\right)+f_{Conv}\left(M_{0}\right)$$
where $f_{Dw}$ refers to the depth-wise convolution operation and ⊙ represents the element-wise multiplication.

## 5. Experiments

This section firstly introduces the experimental settings including the datasets, the network training process, and the evaluation metrics used. Next, we assess the performance of our proposed MG-S2MLPNet using both simulated and real-world HSI datasets and then conduct an ablation study to highlight the impact and effectiveness of different proposed components.

### 5.1. Experimental Settings

**Datasets:** We utilised two distinct HSI datasets—CAVE [53] and KAIST [54]—for our simulation experiments. The CAVE dataset comprises 32 high-quality HSIs, each with a spatial resolution of 512×512 pixels. These images capture a variety of objects and scenes and have good variety for model training. In contrast, the KAIST dataset offers 30 HSIs with significantly larger spatial dimensions of 2704×3376 pixels, providing more detailed visual information. In our experiments, the CAVE dataset was employed exclusively for model training due to its compact image size and diverse content, and we synthesised the snapshot measurements using the released sensing mask in the recent work [51], without using spatial shift. To assess the model’s performance, we selected a subset of 10 cropped scenes from the KAIST dataset for evaluation. These cropped scenes were chosen following same experimental settings as in [22,23,32,33,42,45], ensuring a fair and consistent benchmarking comparison.

In the real-world data experiment, we used five HSIs collected from the TSA-Net work [22] to evaluate the performance of our method. These real scenes were collected using the general CASSI setup with a shift step d = 2 in the disperser. Each of these HSIs consists of 660×660 pixels and 28 spectral bands, capturing a rich and detailed representation of the scene’s spectral properties. For training, we followed the strategy used in [42], using samples extracted from both the CAVE and KAIST datasets. These training samples were selected to match the dimensions of the test data, with patches sized at 660×660×28.

**Implementation Details:** We implemented our proposed MG-S2MLPNet model using the PyTorch framework. The model was trained using the Adam optimizer, configured with hyperparameters $\beta_{1}=0.9$ and $\beta_{2}=0.999$, to control the momentum and smooth the optimization process. We trained the model for a total of 300 epochs and employed a cosine annealing scheduler to adjust the learning rate during training. The initial learning rate was set to $4\times 10^{-4}$, and the batch size was set as 2. To enhance the diversity of the training samples, data augmentations such as random flipping and rotation were applied during training.

**Evaluation Metrics:** The effectiveness of our HSI restoration method will be evaluated using two quantitative metrics: the Peak Signal-to-Noise Ratio (PSNR) and Structural Similarity Index (SSIM). These metrics provide a quantitative assessment of the quality of the restored image, with PSNR measuring the fidelity of the reconstruction by comparing the restored image to the ground truth, while SSIM evaluates the structural consistency and perceptual quality between the two images.

### 5.2. Quantitative Results

We conducted a thorough comparative analysis of our proposed MG-S2MLPNet and several state-of-the-art (SOTA) HSI restoration methods. The SOTA techniques included three model-based approaches: TwIST [35], GAP-TV [20], and DeSCI [17]; four E2E methods: TSA-Net [22], HDNet [23], MST [42], and CST [41]; and eight DUMs: DGSMP [31], GAP-Net [55], ADMM-Net [40], DAUHST [32], PADUT [33], MAUN [56], and RDLUF-L [45]. To ensure a fair comparison, all methods were trained on the same datasets and evaluated under identical conditions. The results, compared using the evaluation metrics PSNR and SSIM and based on 10 simulated scenes, are presented in Table 1. From Table 1, it can be observed both that E2E models significantly outperform traditional model-based methods and that the DUMs usually obtain further performance improvements compared to the E2E methods. Our proposed MG-S2MLPMet surpasses all competing techniques. Specifically, when compared to HDNet [23], MST-L [42], CST-L [41], DAUHST-L [32], PADUT-L [33], MAUN-L [56], and RDLUF-L [45], our proposed model delivers notable improvements in its average PSNR, with gains of 5.15 dB, 4.96 dB, 4.00 dB, 1.76 dB, 1.23 dB, 1.07 dB, and 0.55 dB, respectively. In addition to the superior performance it achieves, our model also boasts lower computational costs and memory overhead, as demonstrated, making it both highly efficient and effective for HSI restoration tasks.

### 5.3. Qualitative Results

**Simulation Results:** We also provide a visualization of the comparison of the proposed MG-S2MLPNet for HSI reconstruction by evaluating 3 out of the 28 spectral channels from Scene 1 and juxtaposing the results against eight SOTA methods. As illustrated in Figure 3, our approach yields reconstructions with visually smoother and cleaner textures, while also maintaining the spatial coherence of homogeneous regions. Moreover, the difference between the ground-truth images and the reconstruction made by our proposed model is much smaller than that of other methods at almost spatial positions. These findings demonstrate the efficacy of our method in producing high-quality HSIs characterised by enhanced texture representations and the preservation of spatial information. Additionally, we evaluated the spectral fidelity of the reconstructed images by comparing the spectral density curves of the restored regions with the ground truth. As presented in Figure 4, our method attained the highest correlation coefficient, underscoring the effectiveness of our mask-guided S2MLP learning strategy in preserving spectral consistency and accuracy.

**Real Results:** To evaluate the reconstruction capabilities of our MG-S2MLPNet on real-world data, we retrained the model using both the CAVE and KAIST datasets, applying a real mask with dimensions of 660×660. For testing, we utilised five real-world scenes from the collected dataset [22]. The reconstruction outcomes for two of these real scenes, as depicted in Figure 5, enable a comparative analysis of our proposed MG-S2MLPNet and seven SOTA methods. Upon closer examination of the visual reconstructions, our model consistently delivers a superior performance, particularly in terms of texture clarity and spatial coherence. Fine details and textures appear more pronounced and distinct in the MG-S2MLPNet results compared to the competing approaches. Additionally, our method demonstrates a better preservation of spatial integrity, avoiding the distortions or blurring that often diminish the quality of the reconstructions produced by the other SOTA techniques. Overall, MG-S2MLPNet shows strong competitiveness, offering more faithful and visually accurate reconstructions.

### 5.4. Ablation Study

To thoroughly analyse the contribution of each component to the performance improvements of our model, we conducted a detailed ablation study. Initially, we used the S2MLP model without the mask modelling branch as the baseline. In this baseline, we varied the number of feature channels (*C*) within the S2MLP module and the S2MLP block number (*B*) to examine their impact on performance. As shown in Table 2, the increased channel and block numbers are capable of improving the model’s reconstruction performance, while higher numbers cause high computational costs. It can be observed from Table 2 that three blocks of S2MLP with 28 spectral and 56 spatial channels achieve an impressive performance with a much lower computational cost, as seen in Table 1. Then, we conducted ablation study using the network configuration with three S2MLP blocks. We incrementally introduced the key components of our framework: the attention-based mask modelling branch (AMMM), the projection loss, the multi-level information fusion module (MIFM), and the multiple prediction heads for deep supervision (DS). Each of these components was added one by one to isolate their individual contributions to the overall enhancement of the model. The comparative results are presented in Table 2, clearly demonstrating that each of the proposed modules contributes positively to performance. The integration of the AMMM refines feature extraction according to the degradation degrees hidden in the sensing mask, while the projection loss enforces consistency in the optical measurement process, leading to more accurate reconstructions. The MIFM further enhances the model by fusing information at various levels of abstraction, improving the depth and richness of the features. Finally, deep supervision (DS) with multiple prediction heads (α=0.2) provides finer control over intermediate representations, allowing the model to refine predictions at different stages of processing. Together, these components work synergistically to significantly elevate the model’s performance beyond its baseline.

To evaluate the effectiveness of S2MLP in comparison to CNN-based architectures and conventional transformer models, we substituted the S2MLP blocks with ResBlock, the Swin Transformer (Swin) [43], and the Spectral Transformer (Spe-Trans) [42]. In this experimental setup, the step size of the CycleFC in S2MLP was set to 11. As indicated in Table 3, S2MLP achieved the highest performance, exceeding SW-MSA and S-MSA by 4.23, 0.65, and 0.77 dB, respectively, with lower computational complexity. These findings underscore that the S2MLP-based model provides a streamlined and efficient approach for achieving superior performance in hyperspectral image reconstruction. Further, we assess the impact of the step-size parameter within the CycleFC layer by testing step sizes of 7, 9, 11, and 13. As shown in Table 3, a step size of 11 results in the highest performance among the configurations examined. Finally, we adjust the hyperparameter α in Equation (Equation 7) to assess its influence on reconstruction performance. The results indicate that variations in α do not significantly impact the overall performance.

To provide insights into the recovered spatial and spectral data both with (w/) and without (w/o) degradation information incorporated, we present difference images that show the discrepancies between the ground-truth data and the reconstructed data across 9 out of 28 spectral bands. Figure 6 illustrates comparative visualizations of Scenes 2 and 5, confirming that reconstruction errors with the AMMM method are significantly reduced across most spatial regions. Additionally, we calculate the PSNR values for each spectral band w/ and w/o the use of ADMM; the resulting comparison curves are illustrated on the right side of Figure 6. These curves also demonstrate that incorporating degradation information (using AMMM) improves PSNR values across all spectral bands. For Scenes 2 and 5, while the PSNR improvements for spectral bands within the 480–500 nm range show a slight degradation, they still exceed an increase of 2 dB.

## 6. Conclusions

In this study, we introduced a novel yet simple S2MLP model designed to effectively tackle HSI reconstruction. Our approach concentrated on leveraging a powerful yet computationally efficient spatial and spectral modelling mechanism by using the CycleFC framework. This integration ensures that our model can capture complex spatial and spectral patterns without imposing a heavy computational burden. Furthermore, we enriched the model’s learning capacity by incorporating a attention-based mask modelling branch, which facilitates the disentangling of representations, allowing the network to better separate and handle different features during the learning process. To further refine the model’s capabilities, we explored multi-level information fusion, where information from different levels of the network is combined. This, coupled with supervision applied to multiple feature layers (deep supervision), strengthens the model’s ability to learn more robust and detailed representations. Extensive experimental evaluations underscored the effectiveness of our proposed approach. The results consistently demonstrate that our model excelled in producing high-quality reconstructions, surpassing SOTA methods in terms of its accuracy and preservation of detail.

## Figures and Tables

**Figure 1 sensors-24-07362-f001:**
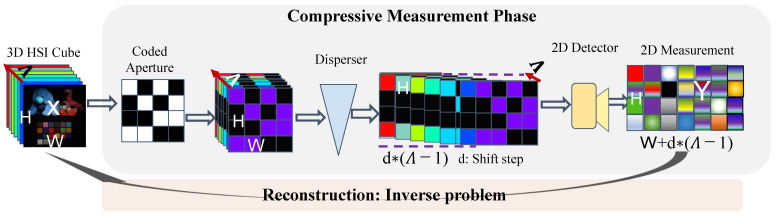
The conceptual scheme of the coding process in a spectral compressive imaging system.

**Figure 2 sensors-24-07362-f002:**
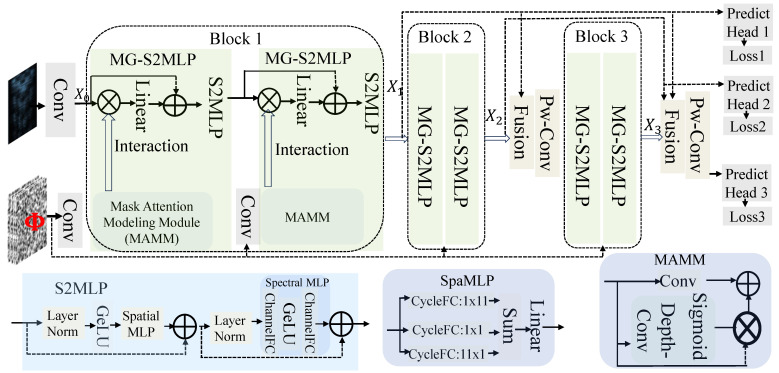
An overview of the proposed MGS2MLP model.

**Figure 3 sensors-24-07362-f003:**
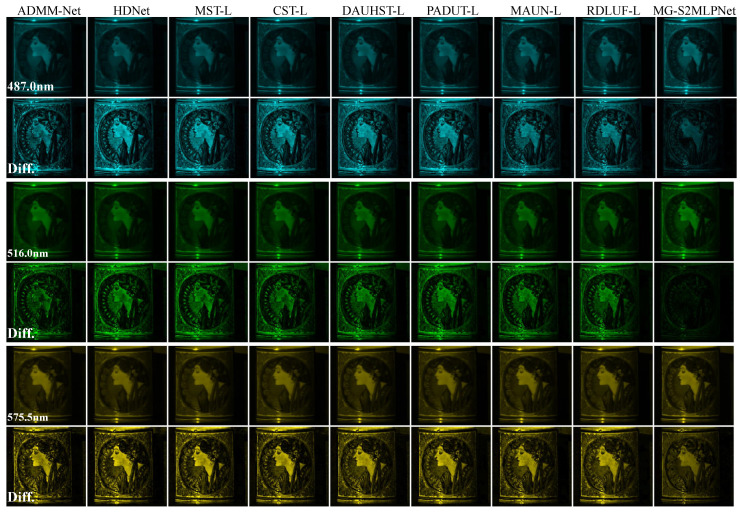
Comparison of visual quality of Scene 1, using 3 (out of 28) spectral channels, and the difference between the ground truth images and those reconstructed with 8 SOTA methods.

**Figure 4 sensors-24-07362-f004:**
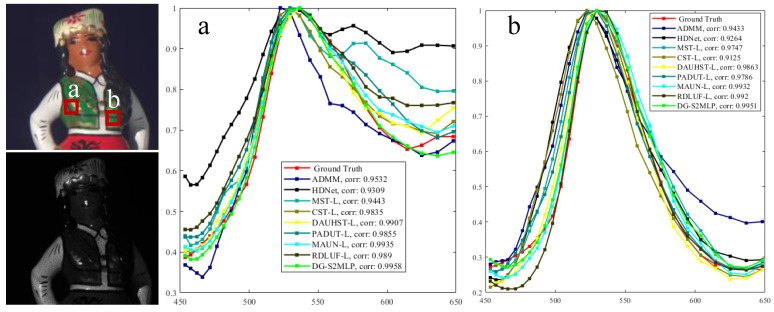
Spectral density curves in ‘a’ and ‘b’ corresponding to two small regions: a and b in Scene 6. The correlation refers to the coincidence degree between the reconstruction and ground-truth curves.

**Figure 5 sensors-24-07362-f005:**
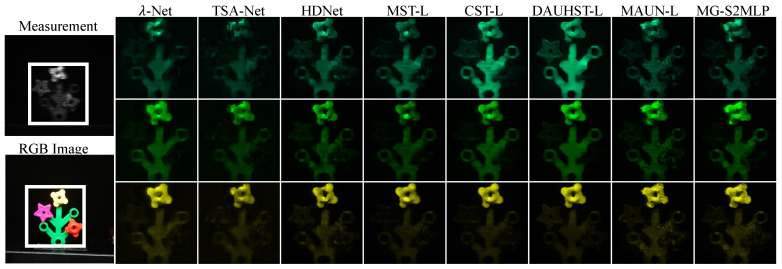
HSI reconstruction results of the proposed MG-S2MLP and seven SOTA methods on the real Scene 1.

**Figure 6 sensors-24-07362-f006:**
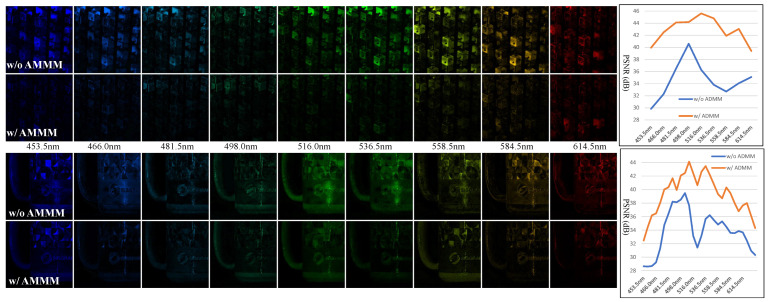
The comparative visualisation of the difference images between the ground-truth data and reconstructed data across 9 out of 28 spectral bands, and the PSNR values of all spectral bands with and without degradation incorporation (AMMM) for Scenes 2 and 5.

**Table 1 sensors-24-07362-t001:** Comparisons with SoTA methods over 10 simulation scenes.

Methods	Params	GFLOPs	s1	s2	s3	s4	s5	s6	s7	s8	s9	s10	Avg
TwIST [35]	-	-	25.16	23.02	21.40	30.19	21.41	20.95	22.20	21.82	22.42	22.67	23.12
			0.700	0.604	0.711	0.851	0.635	0.644	0.643	0.650	0.690	0.569	0.669
GAP-TV [20]	-	-	26.82	22.89	26.31	30.65	23.64	21.85	23.76	21.98	22.63	23.1	24.36
			0.754	0.610	0.802	0.852	0.703	0.663	0.688	0.655	0.682	0.584	0.669
DeSCI [17]	-	-	27.13	23.04	26.62	34.96	23.94	22.38	24.45	22.03	24.56	23.59	25.27
			0.748	0.620	0.818	0.897	0.706	0.683	0.743	0.673	0.732	0.587	0.721
λ-Net [24]	62.64M	117.98	30.10	28.49	27.73	37.01	26.19	28.64	26.47	26.09	27.50	27.13	28.53
			0.849	0.805	0.870	0.934	0.817	0.853	0.806	0.831	0.826	0.816	0.841
TSA-Net [22]	44.25M	110.06	32.03	31.00	32.25	39.19	29.39	31.44	30.32	29.35	30.01	29.59	31.46
			0.892	0.858	0.915	0.953	0.884	0.908	0.878	0.888	0.890	0.874	0.894
DGSMP [31]	3.76M	646.65	33.26	32.09	33.06	40.54	28.86	33.08	30.74	31.55	31.66	31.44	32.63
			0.915	0.898	0.925	0.964	0.882	0.937	0.886	0.923	0.911	0.925	0.917
GAP-Net [55]	4.27M	78.58	33.74	33.26	34.28	41.03	31.44	32.40	32.27	30.46	33.51	30.24	33.26
			0.911	0.900	0.929	0.967	0.919	0.925	0.902	0.905	0.915	0.895	0.917
ADMM-Net [40]	4.27M	78.58	34.12	33.62	35.04	41.15	31.82	32.54	32.42	30.74	33.75	30.68	33.58
			0.918	0.902	0.931	0.966	0.922	0.924	0.896	0.907	0.915	0.895	0.918
HDNet [23]	2.37M	154.76	35.14	35.67	36.03	42.30	32.69	34.46	33.67	32.48	34.89	32.38	34.97
			0.935	0.940	0.943	0.969	0.946	0.952	0.926	0.941	0.942	0.937	0.943
MST-L [42]	2.03M	28.15	35.40	35.87	36.51	42.27	32.77	34.80	33.66	32.67	35.39	32.50	35.18
			0.941	0.944	0.953	0.973	0.947	0.955	0.925	0.948	0.949	0.941	0.948
CST-L [41]	3.00M	40.01	35.96	36.84	38.16	42.44	33.25	35.72	34.86	34.34	36.51	33.09	36.12
			0.949	0.955	0.962	0.975	0.955	0.963	0.944	0.961	0.957	0.945	0.957
DAUHST-L [32]	6.15M	79.50	37.25	39.02	41.05	46.15	35.80	37.08	37.57	35.10	40.02	34.59	38.36
			0.958	0.967	0.971	0.983	0.969	0.970	0.963	0.966	0.970	0.956	0.967
PADUT-L [33]	5.38M	90.46	37.36	40.43	42.38	46.62	36.26	37.27	37.83	35.33	40.86	34.55	38.89
			0.962	0.978	0.979	0.990	0.974	0.974	0.966	0.974	0.978	0.963	0.974
MAUN-L [56]	3.77M	143.83	37.78	40.53	41.88	46.85	36.74	37.78	37.44	36.05	40.54	34.90	39.05
			0.963	0.976	0.973	0.986	0.973	0.974	0.961	0.971	0.973	0.962	0.971
RDLUF [45]	1.81M	115.16	37.94	40.95	**43.25**	**47.83**	37.11	37.47	38.58	35.50	**41.83**	35.23	39.57
			0.966	0.977	0.979	0.990	0.976	0.975	0.969	0.970	0.978	0.962	0.974
MG-S2MLPNet	0.31M	15.12	**39.47**	**42.26**	41.39	45.08	**39.15**	39.86	**38.97**	**37.05**	40.93	**37.05**	**40.12**
			**0.982**	**0.989**	**0.982**	**0.990**	**0.988**	**0.988**	**0.976**	**0.980**	**0.987**	**0.988**	**0.985**

**Table 2 sensors-24-07362-t002:** Break-down of ablation study.

CS2MLP (C = 28, B = 3)				✓				
CS2MLP (C = 56, B = 2)						✓		
CS2MLP (C = 56, B = 4)								✓
CS2MLP (C = 56, B = 3)	✓	✓	✓		✓	✓	✓	✓
MAMM		✓	✓	✓	✓	✓	✓	✓
P-loss			✓	✓	✓	✓	✓	✓
MIFM					✓	✓	✓	✓
DS						✓	✓	✓
PSNR	34.89	39.53	39.66	39.27	39.97	39.10	40.12	40.25
SSIM	0.962	0.983	0.984	0.981	0.985	0.984	0.985	0.985

**Table 3 sensors-24-07362-t003:** Comparisons with other architectures and different step sizes in the CycleFC layer.

	Architectures				Step Sizes in CycleFC				α Value in Equation (Equation 7)		
	ResBlock	SWin	Spe-Trans	S2MLP	7	9	11	13	0.1	0.2	0.5
PSNR	35.89	39.47	39.35	40.12	39.97	40.11	40.12	39.96	40.13	40.12	40.10
SSIM	0.968	0.981	0.980	0.985	0.984	0.985	0.985	0.984	0.985	0.985	0.985

## Data Availability

Data are available in a publicly accessible repository. The datasets used are the CAVE and KAIST datasets. The CAVE dataset is available at https://cave.cs.columbia.edu/repository/Multispectral 4 April 2024, and the Harvard dataset is available at http://dx.doi.org/10.1145/3130800.3130810 5 April 2024.

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
