# Peer review of "Mask-Guided Spatial–Spectral MLP Network for High-Resolution Hyperspectral Image Reconstruction"

_sensors, 2024, doi:10.3390/s24227362_

Round 1

Reviewer 1 Report

Comments and Suggestions for Authors

This article proposes a new deep learning model, called MG-S2MLPNet, for hyperspectral image reconstruction. The method uses a neural network architecture based on MLPs (Multi-Layer Perceptrons) to efficiently capture spatial and spectral information with low computational cost.

The authors have appropriated equations, graphs, and figures. They are well-argued and well-explored in the text.

The text is well-written, although it is a difficult text, but understandable. In this sense, I highlight the detailed description of the Capture Process: Equations (1), (2), and (3) describe step by step how the CASSI system captures the 3D hyperspectral image and compresses it into a 2D measurement. The use of clear notations and explanation of terms (H, W, Λ, Mλ, d, etc.) facilitate the understanding of the process.

On the other hand, some passages can be improved. Therefore, I suggest some observations to improve the article.

The mathematical formalization of the inverse problem brought me a question. Equation (4) formulates the problem of reconstructing the hyperspectral image from the compressed measurement, highlighting the ill-posed nature of the problem due to the sensing matrix Φ. However, considering equation 4, how do the specific characteristics of the sensing matrix Φ, such as its structure and the method of encoding spectral information, influence the quality and efficiency of hyperspectral image reconstruction? Does this influence vary for different types of coding masks?

The authors mention that S2MLP uses three parallel CycleFC layers with different stride sizes (SH x 1, 1 x 1, and 1 x SW) to capture spatial interactions in various receptive fields. However, I understand that it would be interesting to investigate the impact of stride size choice on model performance. It might be interesting for the methodology and potential replication to explore a bit more about how different stride sizes affect S2MLP's ability to capture long-range spatial relationships and how this translates into improvements in reconstruction quality.

It may be that I didn't understand well, but the article did not discuss the impact of network depth on S2MLPNet performance. I suggest adding an analysis of how increasing or decreasing the number of S2MLP blocks affects reconstruction quality and computational cost would be useful in determining the optimal model configuration. Since the authors emphasize the issue of computational cost.

Equations (7) and (8) present the loss function, but there are no details about the optimization process itself. I suggest including a more complete explanation of the optimization algorithm used, how it minimizes the loss function, and the hyperparameters involved.

Author Response

Comment 1: The influence of the HS image reconstruction performance with different sensing matrix Φ.

Response: Thanks very much for pointing this issue.  We added a subsection (2.3) to clarify the employed coded mask in the spectral snapshot compressive system, and also provided some explanations about the used sensing mask after Eq. (4) in the revised manuscript. The SoTA methods generally investigate the effectiveness of the explored HS reconstruction methods under the released sensing mask for fair comparison. The influence of the coded mask to the HS image reconstruction is also a research direction, and some recent works attempt to design/optimize the coded mask to give best reconstruction performance or according to the content of the captured scenes especially in the video compressive imaging systems. We are going to discuss the effectiveness of different sensing mask for the HS image reconstruction in the future work. Thanks.

Comment 2: Different stride sizes (SH x 1, 1 x 1, and 1 x SW) of three parallel CycleFC layerss.

Response: Thanks for helping to improve our paper. We provided the compared results with different stepsizes (7, 9, 11 and 13) in the CycleFC layer in Table 3 of the revised manuscript, and give some explanation in the subsection of the ablation study. Thanks.

Comment 3: The influence of the S2MLP block numbers on the HS image reconstruction performance.

Response: Thanks very much for pointing this issue.  We added the compared results with different S2MLP block numbers (2, 3 and 4) in the bread-down ablation study Table (table 2) of the revised manuscript. The compared results show that three S2MLP blckos can obtain 0.98 dB improvement of the PSNR value to two S2MLP block while four S2MLP blocks get 0.13 dB PSNR improvement to three blocks. We also give some explanation about the comparison in the ablation study section. Thanks a lot.

Comment 4: Regarding to the optimization process and the involved hyperparameter \alpha in loss function.

Response: Thanks for helping to improve our paper. We employed dual loss function for evaluating the errors in both reconstruction and compressive domains. At same times, we also put the prediction heads on the features of different levels of S2MLP blocks for formulating multi-losses. The total loss is used for our model training. In training process, we simply set the Adam as the optimizer without any other parameter turning. In our experiment of the first submission, we set the hyperparameter \alpha I Eq. (7) as 0.2. We give some comparisons with different \alpha values (0.1, 0.2 and 0.5) in Table 3 of the revised manuscript, which shows that there is no large impact to the reconstruction performance by adjusting the hyperparameter \alpha.  Thanks a lot.

Reviewer 2 Report

Comments and Suggestions for Authors

This paper introduces a novel method for hyperspectral image (HSI) reconstruction in Compressive Sensing (CASSI) systems. The approach addresses the high computational overhead of current deep unfolding frameworks by proposing a lightweight MLP-based network that incorporates degradation information (sensing mask) to separate degradation and latent target features. Key contributions include a multi-level fusion module for enhancing information flow, an attention-based mask modeling module for spatial/spectral adaptability, and a dual-domain loss that combines projection and reconstruction losses. The method achieves notable improvements in reconstruction accuracy and efficiency, with reduced memory and computational demands. Below are my comments, questions, and suggestions to improve the clarity and rigor of this study:

  1. The choice to replace iterative optimization stages with an MLP-based network is interesting, and the paper positions this approach as more efficient than previous methods. Could the authors elaborate on the specific advantages of using an MLP in this context, as opposed to other architectures like CNNs or Transformers that are typically used for spectral and spatial feature extraction?

  2. The  Attention-Based Mask Modeling ModuleAttention-Based Mask Modeling module aims to adaptively represent spatial and spectral degradation. It would be helpful if the authors could clarify the mechanism by which this module differentiates between spatial and spectral degradation. How does this approach compare to more traditional attention mechanisms in terms of capturing non-local dependencies and reducing reconstruction error?

  3. The method is presented as adaptable to different CASSI configurations. Could the authors discuss its adaptability to other compressive sensing setups or HSI acquisition systems?

  4. Given the model’s novelty in addressing degradation information, it would be helpful if the authors could explain the interpretability of the reconstructed features. Does the MLP-based model provide insights into which spectral or spatial features are most crucial for accurate HSI reconstruction?

  5. Additionally, I suggest incorporating recent references, such as Hanachi et al. ("Multi-view graph representation learning for hyperspectral image classification with spectral–spatial graph neural networks ") and Ding et al. ("Integrating prototype learning with graph convolution network for effective active hyperspectral image classification"), to further enrich the manuscript.

Author Response

Comment 1: Could the authors elaborate on the specific advantages of using an MLP in this context, as opposed to other architectures like CNNs or Transformers that are typically used for spectral and spatial feature extraction?

Response: Thanks very much for helping to improve our paper. As we presented in ‘Introduction’ section, MLP-like operations offer a promising alternative of the self-attention mechanism in transformers, and potentially excel at capturing non-local similarities and modeling long-range dependencies, which has gained considerable potential applicability in the vision community. Notably, they have delivered comparable performance in various vision tasks while reducing memory usage and computational costs. In this study, we explored a novel Spatial and Spectral MLP (S2MLP) block, designed to effectively capture long-region representations from both non-local spatial positions and spectral bands for HS image reconstruction task. Including the recent research insights, we also added the compared experimental results with CNN-based and Transformer (Self-attention mechanism)-based architecture, and Table 3 of the revised manuscript shows the comparisons. The results indicate that the proposed S2MLP-based model achieved the best reconstruction performance. The explanations about the comparisons are given in the ‘ablation study’ subsection of the revised manuscript. Thanks a lot.

Comment 2: Regarding to the Attention-Based Mask Modeling Module: Attention-Based Mask Modeling module aims to adaptively represent spatial and spectral degradation. It would be helpful if the authors could clarify the mechanism by which this module differentiates between spatial and spectral degradation. How does this approach compare to more traditional attention mechanisms in terms of capturing non-local dependencies and reducing reconstruction error?

Response: Thanks for the expert reviewer’s comment. This study attempts to incorporate the sensing mask information into the deep learning network to disentangle the degradation and the latent target representation. For the mask modeling, we explore an attention-based mask modeling module, which employs very simple convolution/Sigmoid operations instead of the non-local dependency modeling in the modern Transformer. Based on the observation that the sensing masks released by recent spectral snapshot imaging researches manifests spatial correlation in local spatial region but do not have relevant patterns or structures in distant regions. Thus, we employ the (depth-)convolution and sigmoid operations to automatically learn the adaptive degradation knowledge corresponding to each level of the deep learning network. We claim that employment of the transformer block, which can capture the non-local dependencies, is not necessary for modeling mask information. To verify our analysis, we integrate simple convolution and  Swin Transformer block for the mask modeling, causing much high computational cost, while the HS reconstruction performance was little worse than our simple MAMM.  According to our analysis, we did not provide this compared results in the revised manuscript. Thanks.

Comment 3: Could the authors discuss its adaptability to other compressive sensing setups or HSI acquisition systems?

 Response: Thanks for the expert reviewer’s comment. To provide fair comparisons with the SoTA HS reconstruction methods, we employed the widely used compressive sensing setup such as the CASSI setup (Reference 22) for the real experiments, and the proposed imaging setup in the recent work (Reference 52) for simulation experiments. We also provide some detailed explanations in ‘Experimental setting’ section and section 3. Thanks.

Comment 4: Given the model’s novelty in addressing degradation information, it would be helpful if the authors could explain the interpretability of the reconstructed features. Does the MLP-based model provide insights into which spectral or spatial features are most crucial for accurate HSI reconstruction?

Response: Thanks very much for helping to improve our paper. To verify the reconstruction accuracy of the proposed CS2MLP model w/ and w/o the degradation modeling (ADMM), we provided visualization of the deference images between the ground-truth and reconstruction of 9 spectral bands, and also showed the PSNR plots for all spectral bands in Figure 6 of the revised manuscript. From Figure 6, it can be observed that the incorporation of the degradation information (w/ ADMM) can significantly reduce the reconstruction errors at most of spatial positions (the difference images), and improve the PSNR values of all spectral bands (the PSNR plot at the right side of Figure 6). While the PSNR improvements for spectral bands within the 480nm–500nm range show slight degradation, they still exceed an increase of 2dB. Thanks.

Comment 5: Additionally, I suggest incorporating recent references.

Response: Thanks for the expert reviewer’s comment. We added the mentioned references (3 and 7) in the revised manuscript. Thanks a lot.
